# Possibility of Using NO Modulators for Pharmacocorrection of Endothelial Dysfunction After Prenatal Hypoxia

**DOI:** 10.3390/ph18010106

**Published:** 2025-01-16

**Authors:** Igor Belenichev, Olena Popazova, Oleh Yadlovskyi, Nina Bukhtiyarova, Victor Ryzhenko, Sergii Pavlov, Valentyn Oksenych, Oleksandr Kamyshnyi

**Affiliations:** 1Department of Pharmacology and Medical Formulation with Course of Normal Physiology, Zaporizhzhia State Medical and Pharmaceutical University, 69000 Zaporizhzhia, Ukraine; i.belenichev1914@gmail.com; 2Department of Histology, Cytology and Embryology, Zaporizhzhia State Medical and Pharmaceutical University, 69000 Zaporizhzhia, Ukraine; 3Institute of Pharmacology and Toxicology, National Medical Academy of Ukraine, 03057 Kyiv, Ukraine; 4Department of Clinical Laboratory Diagnostics, Zaporizhzhia State Medical and Pharmaceutical University, 69000 Zaporizhzhia, Ukraine; 5Department of Medical and Pharmaceutical Informatics and Advanced Technologies, Zaporizhzhia State Medical University, 69000 Zaporizhzhia, Ukraine; 6Faculty of Medicine, University of Bergen, 5020 Bergen, Norway; 7Department of Microbiology, Virology and Immunology, I. Horbachevsky Ternopil State Medical University, 46001 Ternopil, Ukraine; kamyshnyi_om@tdmu.edu.ua

**Keywords:** prenatal hypoxia, cardioprotective, endothelioprotection, NO system, angiolin, L-arginine, thiotriazoline, Mildronate

## Abstract

Prenatal hypoxia (PH) is a key factor in the development of long-term cardiovascular disorders, which are caused by various mechanisms of endothelial dysfunction (ED), including those associated with NO deficiency. This emphasizes the potential of therapeutic agents with NO modulator properties, such as Thiotriazoline, Angiolin, Mildronate, and L-arginine, in the treatment of PH. **Methods:** Pregnant female rats were given a daily intraperitoneal dose of 50 mg/kg of sodium nitrite starting on the 16th day of pregnancy. A control group of pregnant rats received saline instead. The resulting offspring were divided into the following groups: Group 1—intact rats; Group 2—rat pups subjected to prenatal hypoxia (PH) and treated daily with physiological saline; and Groups 3 to 6—rat pups exposed to prenatal hypoxia and treated daily from the 1st to the 30th day after birth. Levels of sEPCR, Tie2 tyrosine kinase, VEGF-B, SOD1/Cu-Zn SOD, GPX4, and GPX1 in the heart’s cytosolic homogenate were assessed using ELISA. The expression of VEGF and VEGF-B mRNA was analyzed via real-time polymerase chain reaction, and the nuclear area of myocardial microvessel endothelial cells was evaluated morphometrically. **Results:** We have shown that only two representatives of this group—Angiolin and Thiotriazoline—are able to exert full effect on the indices of endothelial dysfunction after PH to decrease sEPCR, increase Tie-2, VEGF-B and VEGF-B mRNA, Cu/ZnSOD, and GPX in myocardial cytosol, and increase the area of endotheliocyte nuclei in 1- and 2-month-old rats in comparison with the control. **Conclusions:** Our results experimentally substantiate the necessity of early postnatal cardio- and endothelioprotection using NO modulators, taking into account the role of NO-dependent mechanisms in the pathogenesis of cardiovascular system disorders in neonates after PH.

## 1. Introduction

The global progression of cardiovascular disease is influenced by a myriad of factors, including lifestyle choices, dietary habits, unhealthy behaviors, and socioeconomic shocks, but also by suboptimal intrauterine conditions. Numerous studies have demonstrated that cardiovascular dysfunction in adulthood can be programmed during pregnancy, particularly as a result of poor maternal nutrition, alcohol consumption, substance abuse, chemical exposure, and stress. Among the various complications associated with pregnancies worldwide, fetal hypoxia emerges as one of the most prevalent issues, significantly impacting long-term cardiovascular health [1,2,3].

Prenatal hypoxia (PH) is associated with asymmetric fetal growth restriction, leading to hypertrophic growth of the heart and aorta, altered cardiac function, and sympathetic hyperinnervation of peripheral resistive arteries in newborns. In adulthood, the effects of prenatal hypoxia extend to an increased risk of hypertension, coronary heart disease, ischemic heart disease, heart failure, and metabolic syndrome, as well as heightened susceptibility to ischemic injury. These findings underscore the critical importance of addressing hypoxic conditions during pregnancy to mitigate long-term cardiovascular risks [4,5].

The presence of endothelial dysfunction mechanisms was revealed in the pathology of the cardiovascular system after prenatal hypoxia. Clinical manifestations of functional state disturbance and cardiovascular system maladaptation after prenatal hypoxia directly correlated with signs of endothelial dysfunction (changes in endothelin-1, nitric oxide (NO), vascular endothelial growth factor (VEGF) production, circulating desquamated endotheliocytes) both in newborns and at older ages [6,7,8].

Nitrogen monoxide (NO) system disorders play a certain role in the formation of endothelial dysfunction and cardiovascular pathology, including after prenatal hypoxia. Studies have revealed that prenatal hypoxia can change both the production and bioavailability of NO. During prenatal hypoxia, increased concentrations of superoxyradicals and other reactive oxygen species can lead to oxidative modification of NO and convert it to peroxynitrite, which negatively affects fetal organs [5,9,10].

Hypoxia decreases endothelial nitric oxide synthase (eNOS) expression and can alter its enzymatic activity through various post-translational modifications. In conditions of hypoxia coupled with L-arginine deficiency, eNOS may generate superoxyradicals instead of NO. Such abnormalities in eNOS function are thought to be a major cause of endothelial dysfunction observed in cardiovascular disease [8,11].

Given that prenatal hypoxia exerts both immediate and long-term effects on cardiovascular development, it is essential to explore novel molecular and biochemical markers that reflect the hypoxic impact on the cardiovascular system. Additionally, the development of therapeutic agents targeting these effects is critical. The molecular basis of vascular endothelial dysfunction is a complex and not fully understood problem. In this regard, the “eNOS-L-arginine-NO” system may soon play a key role in the pharmacological correction of endothelial dysfunction. Currently, there are no drugs with specific endothelioprotective activity. However, considering the role of NO in the development of endothelial dysfunction, positive modulators of NO synthesis have attracted the attention of researchers as potential agents for cardio- and endothelioprotection following prenatal hypoxia [12,13,14]. Pharmacologically, the level of NO and its bioavailability can be increased through (1) stimulation of NO synthesis, for example, through therapy with L-arginine (a substrate for eNOS), tetrahydrobiopterin (a cofactor for eNOS), or metabolic cytoprotectors that increase the amount of gamma-butyrobetaine (such as Mildronate or trimetazidine); (2) direct protection of NO from reactive oxygen species (ROS) using thiol antioxidants (e.g., Thiotriazoline, Angiolin); (3) positive modulation of eNOS activity [15]. We observed promising experimental outcomes with the use of nitric oxide (NO) modulators, including L-arginine, Thiotriazoline, Angiolin, and Mildronate, following instances of prenatal hypoxia [12,16]. These agents are also described in other studies, which show that cytoprotective, antioxidative, and anti-ischemic effects of these agents are associated with a positive effect on the nitric oxide monoxide system, and influence the synthesis, bioavailability or transport of this messenger.

Thus, Thiotriazoline (tiazotic acid) is a scavenger of reactive oxygen and nitrogen forms, protects NO from chemical modification and transformation into peroxynitrite, and exhibits cardioprotective, membrane-stabilizing, and anti-ischemic properties. Thiotriazoline has low toxicity at different routes of administration to four species of animals, belongs to the V class of toxicity (practically non-toxic substances) and does not show general toxic, teratogenic, embryotoxic, mutagenic, and carcinogenic actions. It has been used in clinical practice for more than 20 years [17].

Angiolin (3-methyl-1,2,4-triazolyl-5-thioacetate (S)-2,6-diaminohexanoic acid) is a structural analog of Thiotriazoline that regulates the concentration of reactive oxygen species (ROS), protects NO from conversion into peroxynitrite, and modulates the expression of eNOS and vascular endothelial growth factor (VEGF) during ischemia and hypoxia. It exhibits antioxidative, neuroprotective, endothelial-protective, cardioprotective, and anti-ischemic properties. “Angiolin” belongs to the V class of toxicity (practically non-toxic substances)—LD_50_ at parenteral administration: rats—7667 mg/kg; mice—9000 mg/kg; LD_50_ at intragastric administration—rats—15,000 mg/kg; mice—10,309 mg/kg—and does not show general toxic, teratogenic, embryotoxic, mutagenic, and carcinogenic actions. After the permission of the State Expert Center of the Ministry of Health of Ukraine, it successfully passed the first phase of clinical trials [18].

Mildronate affects NO synthesis by increasing the level of gamma-butyrobetaine, and it exhibits cardioprotective and anti-ischemic properties [18,19].

L-arginine, a precursor of NO synthesis, exhibits membrane-stabilizing, cardioprotective, and anti-ischemic properties [20].

Purpose of this study: to perform a comparative evaluation of the effects of NO modulators (L-arginine, Thiotriazoline, angiotensin, and Mildronate) on various markers of endothelial dysfunction (sEPCR, Tie-2, VEGF-B, endothelial cell nuclear density) and the antioxidant system (Cu/ZnSOD, GPX1, GPX4) following experimental PH and to justify further investigation into the cardio- and endothelioprotective effects of the most promising pharmacological agent.

## 2. Results

Prenatal hypoxia (PH) modeling leads to changes in the concentration of various proteins in the heart cytosol of experimental animals, which may indicate the development of endothelial dysfunction (Table 1 and Table 2). We observed a significant increase in the concentration of the soluble form of the endothelial protein C receptor (sEPCR), rising by 1.92 times at 1 month of life and by 2.14 times at 2 months of life compared to the intact group of the corresponding age (*p* ≤ 0.05). Additionally, we found a significant decrease in the tyrosine kinase receptor Tie-2, by 42.3% at 1 month of life and by 37.9% at 2 months. The concentration of vascular endothelial growth factor B (VEGF-B) was also significantly reduced after PH, decreasing by 28.1% and 35.2% at 1 and 2 months, respectively, in experimental animals. We also found a decrease in VEGF mRNA expression by 2.9 times and a decrease in VEGF-B mRNA expression by 5.8 times in the hearts of 1-month-old rats compared to the group of healthy 1-month-old animals. In the hearts of 2-month-old rats, the reduction in VEGF and VEGF-B mRNA expression was even more pronounced compared to the group of healthy 1-month-old rats, with reductions of 3.3 and 6.4 times, respectively. Additionally, we observed that prenatal hypoxia (PH) leads to a greater suppression of VEGF-B mRNA (Table 3 and Table 4). Evidence of endothelial dysfunction in the myocardial microvessels was shown by a decrease in the area of endothelial cell nuclei, which was reduced by 42.6% in 1-month-old rats and 43.4% in 2-month-old rats (Table 5). Furthermore, we identified a reduction in the expression of antioxidant enzymes, which play a crucial role in limiting the damaging effects of oxidative stress intermediates—such as superoxide radicals, hydroperoxides, and lipid peroxides. In the cytosol of rat hearts after PH, a significant decrease in the concentration of the Cu/Zn-dependent isoform of superoxide dismutase (Cu/ZnSOD) by 27.6% (at 1 month of life) and by 31.6% (at 2 months of life) was observed. A reduction in the concentration of glutathione peroxidase 4 (phospholipid hydroperoxidase) (GPX4) was also found at 1 and 2 months of life, by 49.5% and 47.8%, respectively. The concentration of glutathione peroxidase 1 (GPX1) also decreased, by 51.2% at 1 month of life and by 54.3% at 2 months.

Course administration of drugs that are modulators of the nitric oxide system for 30 days immediately after birth leads to varying degrees of normalization in the expression of these proteins (sEPCR, Tie-2, VEGF-B, Cu/ZnSOD, GPX) (Table 1 and Table 2). The administration of L-arginine resulted in a significant reduction in sEPCR by 12.0% immediately after discontinuation of the drug and by 22.4% one month after the end of L-arginine treatment, indicating a lasting effect. L-arginine administration significantly increased the concentration of Tie-2 by 1.4 times in the cytosol of experimental animals both immediately after discontinuation of the drug and one month after the end of the treatment course. However, L-arginine administration did not affect the concentration and expression of VEGF-B and Cu/ZnSOD in the heart cytosol of experimental animals. L-arginine administration resulted in a significant increase in endotheliocyte nuclear cavity in 28% of post-PH myocardial microvessels one month after administration of the drug (Table 5).

The introduction of L-arginine led to a significant increase in GPX4 expression immediately after administration, while GPX1 expression increased a month after the course ended. Thiotriazoline significantly reduced cEPCR levels in the cytosol of the hearts of rats after PH at both observation periods (1 and 2 months of life of experimental animals) by 22.4% and 29.0%, respectively. The course administration of Thiotriazoline resulted in a significant increase in Tie-2 by 24.5% and 39.0% for the respective observation periods (1 and 2 months after PH). Additionally, Thiotriazoline led to a significant increase in VEGF-B by 14.6% and 19.6% for the respective observation periods. Thiotriazoline increased the expression of VEGF mRNA and VEGF-B mRNA by 2.61 and 6.8 times, respectively, in the hearts of 1-month-old rats after prenatal hypoxia (PH), and by 3.9 and 12.8 times, respectively, in the hearts of 2-month-old rats after PH (Table 3 and Table 4). Thiotriazoline significantly elevated the expression of antioxidant enzymes in the cytosol of the myocardium of experimental animals—Cu/ZnSOD by 22.5% and 25.3%, GPX1 by 83.8% and 200%, and GPX4 by 68.7% and 87.7% for the respective observation periods after the drug administration. As can be seen, Thiotriazoline has a greater impact on GPX1, which is consistent with its previously established antioxidant properties. Thiotriazoline demonstrated direct endothelial-protective properties, significantly increasing the nuclear area of endothelial cells in the myocardium of 1- and 2-month-old rats after prenatal hypoxia (PH) by 40.7% and 62%, respectively, compared to the untreated group.

It is noteworthy that the measured indicators in the group of animals with PH receiving Thiotriazoline did not significantly differ from those in the group of animals born after a normally progressing pregnancy. Angiolin demonstrated the most pronounced therapeutic effect among all the studied agents (Table 1 and Table 2). Specifically, Angiolin significantly reduced cEPCR levels in the cytosol of the hearts of rats after PH at both observation periods—immediately after the course of administration and 1 month after its cessation (1 and 2 months of the experimental animals’ life)—by 34.7% and 53.3%, respectively.

It is important to note that cEPCR levels in the cytosol of the myocardium of 2-month-old animals with PH after receiving Angiolin were comparable to those of animals born after a physiologically normal pregnancy. The course administration of Angiolin led to a significant increase in Tie-2 by 60.7% and 62.8% for the respective observation periods (1 and 2 months after PH). cEPCR indices in myocardial cytosol of 2-month-old animals after PH receiving Angiolin were at the level of animals born after physiologically normal pregnancy.

The administration of Angiolin also led to a significant increase in VEGF-B by 48.9% and 67.0% for the respective observation periods. It is important to note that the concentration of VEGF-B in the cytosol of the myocardium of 1- and 2-month-old rats after PH was significantly higher than that of age-matched animals born after a physiologically normal pregnancy. Angiolin increased the expression of VEGF mRNA and VEGF-B mRNA by 5.7 and 12.5 times, respectively, in the hearts of 1-month-old rats after prenatal hypoxia (PH) and by 6.2 and 19 times in the hearts of 2-month-old rats after PH (Table 3 and Table 4). Among all the pharmacological agents tested, Angiolin demonstrated the most pronounced direct endothelial-protective properties, significantly increasing the nuclear area of endothelial cells in the myocardium of 1- and 2-month-old rats after PH by 60.8% and 70%, respectively, compared to the untreated group. Morphometric parameters of endothelial cell nuclei in animals treated with Angiolin after PH were comparable to those of healthy rats.

The use of Angiolin also resulted in an increase in the expression of antioxidant enzymes—Cu/ZnSOD by 25.5% and 41.2%, GPX1 by 92.8% and 130.1%, and GPX4 by 83.6% and 105.8% for the respective observation periods after the drug administration—indicating a significant antioxidant mechanism of action for the drug.

Mildronate, when administered in a course after PH, had the least pronounced effect compared to the other studied drugs (Table 1 and Table 2). We observed a significant change in the group receiving Mildronate compared to the control group in the levels of Tie-2 and GPX4 immediately after a one-month course of administration, as well as a significant change compared to the control group in the levels of VEGF-B, Tie-2, GPX1, and GPX4 one month after the course of the drug.

## 3. Discussion

The modeling of PH through the introduction of sodium nitrite to pregnant females leads to hemic hypoxia due to the formation of methemoglobin. This hypoxia is accompanied by tissue hypoxia caused by the uncoupling of oxidation and phosphorylation processes. The disruption of blood oxygen transport in pregnant female rats results in impaired uteroplacental blood flow and oxygen starvation of the fetus or embryo [21]. Administration of sodium nitrite at a dose of 50 mg/kg leads to hypoxia of medium severity in adult individuals, according to the criteria proposed by N.F. Ivanitskaya [22].

The modeling of PH leads to the development of postnatal heart defects. In both newborns and adult animals, our model allows for the assessment of the physiological development of offspring and the effectiveness of experimental cardioprotective therapy following prenatal hypoxia (PH). The administration of sodium nitrite to pregnant rats results in increased methemoglobin levels [23], specifically, hypoxic damage to the target organs of the fetus. Our previous studies have shown that modeling chronic hypoxia with sodium nitrite leads to persistent ECG abnormalities, reduced myocardial contractility, and sinus node dysfunction [24], focal dystrophy, as confirmed by an increase in the concentration of a highly sensitive marker of myocardial remodeling and the risk of heart failure, ST2 [16].

We also revealed a significant impairment of the myocardial nitriergic system in rats after prenatal hypoxia (PH)—an imbalance in the ratio of eNOS/iNOS expression against the background of NO deficiency and increased nitrotyrosine levels [12]. This suggests impaired cardiac tolerance to ischemia/reperfusion and damage to endothelial-dependent mechanisms of vasodilation/vasoconstriction, and may further contribute to the development of endothelial dysfunction following intrauterine hypoxia. Endothelial dysfunction after prenatal hypoxia develops against a background of HIF-1α deficiency (a factor that activates eNOS expression through serine residue phosphorylation) and nitrosative stress, which also leads to HSP70 deficiency, glutathione system depletion, reduced NO bioavailability, and suppression of gene transcription by cytotoxic NO products [12,16,25].

Our research findings also confirm that this model of PH leads to pathological changes in the cardiovascular system of newborns and the development of endothelial dysfunction. EPCR levels increase in endothelial cells during post-ischemic neovascularization. It is important to note that the exogenous addition of NO significantly enhanced the formation of endothelial angiogenic sprouts from aortic rings and primary endothelial cells isolated from mice with a PAR1 mutation. Thus, maintaining NO bioavailability during angiogenic processes is a primary function of EPCR-PAR1 endothelial signaling [26,27].

The release of EPCR from the endothelium often leads to the formation of the soluble form of EPCR (sEPCR). It was found that SS mice had higher levels of soluble EPCR (sEPCR) in plasma compared to their AA counterparts. The endothelial protein C receptor (EPCR) plays a crucial role in the anticoagulant and anti-inflammatory effects of the protein C pathway, whereas its soluble form (sEPCR) exhibits opposing properties. High levels of sEPCR in plasma and tissues have been observed in individuals with the A3 haplotype of the PROCR gene, the EPCR gene. Elevated levels of sEPCR in plasma have also recently been reported in women with preeclampsia (PE), a multisystem syndrome involving inflammation, endothelial dysfunction, and thrombosis [28].

Tie2 plays an important role in providing barrier protection to prevent excessive vascular permeability and maintains an antithrombotic surface to improve blood circulation. It remains activated throughout the healthy vascular system of an adult due to the continuous secretion of angiopoietin-1 from perivascular cells and platelets, promoting endothelial stability by inhibiting the inflammatory NF-κB [29]. In animal models simulating critical illness, Tie2 levels in organs are temporarily reduced. The functional consequences of these reduced Tie2 levels for microvascular endothelial behavior are associated with increased microvascular inflammation [30]. It has been shown that mice with null Tie-2 exhibit severe vascular damage and cardiac abnormalities, leading to embryonic lethality, as Tie-2 is essential for supporting the development and stabilization of fetoplacental vessels and regulating NO production [31]. Data have been obtained demonstrating the potential of activating Tie2 with a pharmacological agent, leading to a reduction in the thromboinflammatory state of the endothelium in COVID-19 [29]. VEGF-B is a powerful survival factor for various cell types, inhibiting apoptosis by suppressing the expression of apoptosis-related proteins and genes, and is crucial for the survival of blood vessels; however, it does not induce blood vessel growth. Pharmacological modulation of VEGF-B results in a strong cytoprotective and anti-apoptotic effect without triggering general angiogenic activity [32]. The heart expresses a high level of VEGF-B, which exerts a strong anti-apoptotic effect on cardiomyocytes by suppressing the expression of pro-apoptotic genes (BMF, BAD, BID, BAX, CASP9, DCN, TP53INP1, TNF). VEGF-B induces several antioxidant genes (GPX1, GPX4, SOD-1, SOD-2, etc.) and suppresses genes responsible for oxidative stress. VEGF-B reduces endothelial cholesterol content by inhibiting the recirculation of low-density lipoprotein receptors, influences uptake, and increases the utilization of fatty acids by the myocardium for energy production [33]. Our results are consistent with other studies presented in a review [5]; these studies demonstrated that prenatal hypoxia leads to endothelial dysfunction, reduced NO production, decreased expression and concentration of VEGF, and thickening and deposition of fibrils in the intima, as well as migration and proliferation of smooth muscle cells into the intima of myocardial vessels. The morphological changes in the heart are mediated by endothelial dysfunction. Suppression of VEGF-B expression results in mitochondrial dysfunction, metabolic disorders, and an increased risk of heart failure development [34]. A possible reason for the suppression of VEGF-B mRNA expression during prenatal hypoxia may be an excess of reactive oxygen species (ROS) [35].

Preclinical studies have demonstrated the therapeutic potential of VEGF-B in revascularizing ischemic myocardium by modulating endothelial cell proliferation and migration [36]. It has been established that VEGF-B primarily interacts with Flt-1 (vascular endothelial growth factor receptor) and sFlt-1 (soluble vascular endothelial growth factor receptor-2) and inhibits vascular endothelial dysfunction in preeclampsia. Administration of a recombinant VEGF-B preparation to rodents with experimental preeclampsia restored the angiogenic environment in plasma, normalized blood pressure, and reduced the severity of ischemia [37]. VEGF is indeed an attractive target for our NO modulators (L-arginine, Thiotriazoline and Angiolin), as a NO-dependent mechanistic regulation of VEGF expression has been described [38]. The reduced expression of the main antioxidants, which we have found in the previously detected increase in nitrotyrosine in the myocardium of 1- and 2-month-old rats following PH [12], shows a significant activation of oxidative stress after PH. Oxidative stress in the fetal heart and vasculature underlies the mechanism by which prenatal hypoxia programs cardiovascular pathology and endothelial dysfunction later in life [4]. Our present study and previously published results are not contradicted by other investigators who have shown that PH contributed to aortic thickening with enhanced nitrotyrosine staining and increased expression of cardiac HSP70, as well as marked impairment of NO-dependent relaxation in arteries and increased myocardial contractility with sympathetic dominance [39].

GPX-4 is most important for cellular protection under oxidative stress, directly reducing phospholipid hydroperoxides, even when incorporated into membranes and lipoproteins. GPX-4 can also restore fatty acid hydroperoxide, cholesterol hydroperoxide, and thymine hydroperoxide. It plays a key role in protecting cells from oxidative damage by preventing membrane lipid peroxidation. GPX-4 is required to prevent cells from ferroptosis, non-apoptotic cell death resulting from iron-dependent accumulation of lipid reactive oxygen species [40,41]. GPx4 is required to prevent the death of mitochondrial cells by mediating the reduction in cardiolipin hydroperoxides. GPx4 is involved in the direct detoxification of lipid peroxides in the cell membrane and is an inhibitor of ferroptosis induced by lipid peroxidation. The cytosolic isoform of GPx4 plays a key role in inhibiting ferroptosis in somatic cells, while the mitochondrial isoform of GPx4 (mGPx4) may play a role in reducing the risk of mitochondrial dysfunction [42]. It has been discovered for the first time that PH can lead to ferroptosis in human trophoblast cells, which may subsequently cause miscarriage. This underscores the importance of GPX-4 [43]. GPx-1 is an intracellular antioxidant enzyme that enzymatically reduces H2O2 to H2O to limit its harmful effects, as well as regulates H2O2-dependent signaling mechanisms mediated by growth factors, mitochondrial function, and the maintenance of normal thiol redox balance. Our findings indicate that the decreased expression of GPx-1 in the hearts of rats after PH may be associated with an excess of cytotoxic forms of NO in the context of high iNOS expression [44], as we found in a previous study [12]. GPx-1 plays an important role in maintaining endothelial function and NO bioavailability [44] GPx-1 deficiency leads to marked vasoconstriction and forms endothelial dysfunction [45].

SODs are generally classified into four groups: manganese SOD (MnSOD), copper–zinc SOD (Cu/ZnSOD), iron SOD (FeSOD) and nickel SOD (NiSOD). Cu/ZnSOD and MnSOD localize in the cytoplasm, serve as the major radical scavengers in the intracellular environment, and have attracted much attention because of their physiological function and therapeutic potential [46]. Our studies showing a low concentration of Cu/ZnSOD in the rat cytosol after PH are supported by other studies showing that PH reduces Cu/ZnSOD expression at both transcriptional and post-translational levels. In addition, PH decreases Cu/ZnSOD activity and may be a cause of subsequent cardiovascular disease [47] and endothelial dysfunction [48]. There is strong evidence of a proven link between reduced activity of antioxidant enzymes and the occurrence of adverse pregnancy outcomes, as oxidative stress has a deleterious effect on maternal physiology, pregnancy, and fetal development, impairing placental function and impairing oxygen and nutrient delivery to the developing fetus and contributing to cardiovascular disorders, in particular cardiomyopathy and endothelial dysfunction [49]. Positive modulators of NO, by increasing physiologic concentrations of this messenger, participate in the mechanisms of the S-nitrosation of a cysteine residue and regulate post-translational modification of various proteins, including eNOS [50]. All of this, as well as our previous research [11,12,16], allowed us to justify the use of positive NO modulators in the experimental therapy of cardiovascular complications after PH. Pharmacological agents that elevate nitric oxide (NO) levels and extend its activity can stimulate NO-dependent mechanisms involved in endothelial growth. These agents regulate the expression of vascular endothelial growth factor (VEGF) family proteins, including placental growth factor (PGF), angiopoietins (ANG-1 and ANG-2), and their soluble receptors (sFLT-1 and sTIE-2). Additionally, they promote NO-dependent expression of proangiogenic factors such as VEGF-A, cardioprotective VEGF-B, and PGF, while mitigating NO-dependent expression of endothelial adhesion molecules and proinflammatory cytokines. The highest activity in this study was demonstrated by Angiolin (S)-2,6-diaminohexanoic acid 3-methyl-1,2,4-triazolyl-5-thioacetate, which has NO-scavenging properties, in which fragments of the chemical structure of the molecule take part. Angiolin can form nitrothiols and increase NO bioavailability. The interactions between the Angiolin molecule and NO can be realized by electron transfer from the higher occupied molecular orbital of the “spin trap” to the lower unoccupied molecular orbital of the nitrogen monoxide radical to form a more stable complex compound (Figure 1).

Angiolin normalizes eNOS/iNOS expression. In studies on the model of cerebral ischemia in rats, the endothelioprotective activity of Angiolin was also demonstrated, i.e., an increase in the density of endotheliocytes of muscle-type vessels and microcirculatory channel, an increase in the density of proliferating endotheliocytes, and an increase in the expression of vascular endothelial growth factor (VEGF) and receptor-binding coefficient [11,18]. There is evidence that VEGF enhances the regulation of the enzyme ecNOS and induces a biphasic stimulation of endothelial NO production [52]; this suggests a possible VEGF-mediated expression of eNOS under the action of Angiolin. Angiolin can influence the expression of endotheliotropic factors and antioxidative components through the influence on the thiol–disudyphide system by increasing the level of glutathione and regulating post-translational mechanisms. There is evidence of positive effects on Cu/ZnSOD, GPX1, and GPX4 activity in the cytosol of rat myocardium and brains during cardiac or cerebral ischemia. This may be due to the interruption of NO-dependent mechanisms of suppression of the expression of these enzymes [11,51].

Thiotriazoline, a drug registered in many countries as a methiabolitotropic cardioprotective agent, also exhibits NO scavenger properties, but more moderate effects on eNOS expression in cardiocytes under conditions of myocardial ischemia. Thiotriazoline can increase the endothelioprotective properties of L-arginine by increasing NO bioavailability. Thiotriazoline increases NO bioavailability, preventing its interaction with ROS and conversion into peroxynitrite. This protection is realized due to the co-storage of reduced thiols. Thiotriazoline itself can be a carrier of NO, forming stable S-nitrosyl complexes with it. Thiotriazoline inactivates ROS due to the strong reductive properties of the thiol group (Figure 2). By regulating the ROS level, Thiotriazoline can prevent the inactivation of enzymes, including eNOS, as well as influence red/oxi-dependent mechanisms of gene expression regulation (Figure 3) [15,53,54]. Thiotriazoline exhibits antioxidant properties; in many studies, its ability to reduce the formation of oxidative and nitrosative stress end products and increase the activity of Cu/ZnSOD, GPX1, and GPX4 in the liver, heart, and brain of animals with various experimental pathologies has been established [53].

**Figure 2 pharmaceuticals-18-00106-f002:**
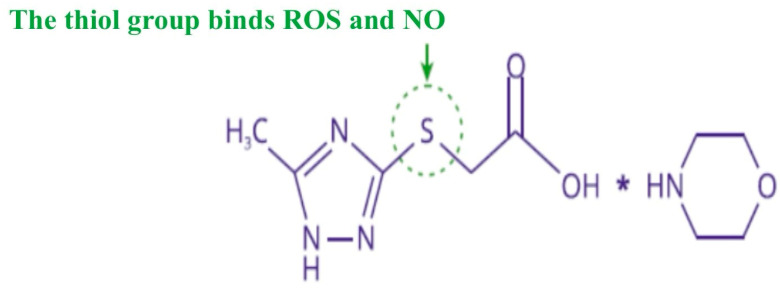
The thiol group binds ROS and NO. The antioxidant properties of Thiotriazoline, and especially its ability to scavenge ROS and cytotoxic forms of NO, can be explained by the reactivity of the sulfur atom in the morpholino-thiazotate molecule. In the presence of peroxynitrite, the sulfur atom of the thiazotate anion interacts with the positively charged nitrogen atom, forming the corresponding adduct. This adduct then releases a hydroxyl anion, leading to the formation of an S-nitro derivative, which upon hydrolysis gives a sulfoxide and a nitrite anion.

**Figure 3 pharmaceuticals-18-00106-f003:**
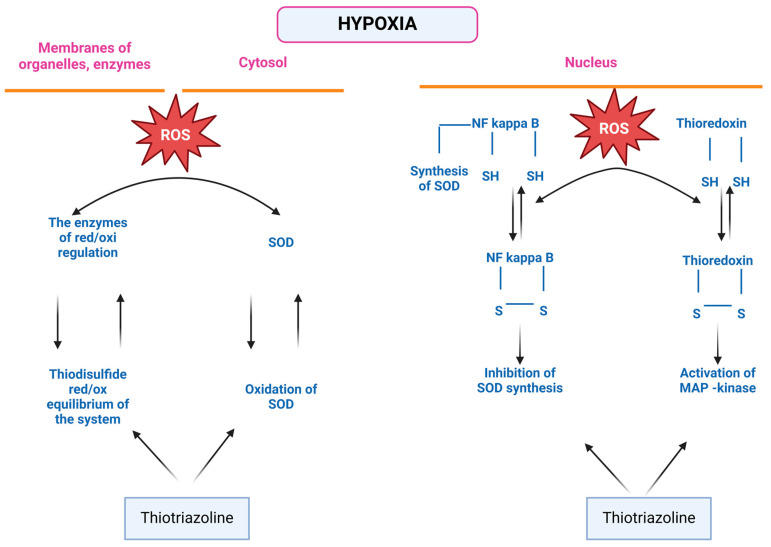
The effect of Thiotriazoline on red/oxi-dependent mechanisms of gene expression. Thiotriazoline prevents the oxidation of cysteine and the formation of cysteine sulfoxide, and inhibits the formation of nitrotyrosine. Based on this, Thiotriazoline prevents the irreversible inactivation of the transcription factor NF-kappa B, protecting the cysteine residues sensitive to ROS—Cys 252, Cys 154, and Cys 61—in its DNA-binding domains (Figure 2). Moreover, Thiotriazoline may participate in the reduction of these groups during reversible inactivation, taking on the role of Redox Factor-1. By inhibiting the oxidative inactivation of the NF-kappa B transcription factor under conditions of excess ROS, Thiotriazoline likely enhances the activation of redox-sensitive gene expression, which is necessary to protect cells from the toxic effects of oxidative stress. Among these genes are those responsible for the synthesis of superoxide dismutase. The protective effect of Thiotriazoline on the sulfhydryl groups of cysteine and methionine fragments of protein molecules has been studied. Thiotriazoline competes with these structures for the superoxide radical, thereby preventing both reversible and irreversible modification. By inhibiting reversible modification, the formation of disulfide bonds (-S-S-) in cysteine regions is prevented. More significant, in terms of efficacy, is Thiotriazoline’s action against the irreversible modification of sulfhydryl groups in several protein molecules under the influence of ROS. Thiotriazoline inhibits the formation of irreversible sulfoxides and sulfonic groups in proteins, which are further prone to oxidation. By exerting an inhibitory effect on the irreversible oxidative modification of sulfhydryl groups in cysteine fragments of protein molecules, Thiotriazoline normalizes redox regulation shifts under oxidative stress conditions. Primarily, Thiotriazoline prevents the disruption of the thiol–disulfide system balance during ROS hyperproduction, ensuring functions such as cellular signal transmission through receptor–ion channel complexes, maintaining the activity of proteins, enzymes, and transcription factors, and the integrity of cell membranes.

Mildronate (3-(2,2,2,2-trimethylhydrazine) propionate) reversibly blocks gamma-butyrobetaine hydroxylase, which catalyzes the conversion of gamma-butyrobetaine into carnitine and thereby significantly inhibits the entry of carnitine, which provides transport of fatty acids across the membrane into the cells of muscle tissue. This effect of Mildronate is accompanied by a decrease in the carnitine-dependent oxidation of free fatty acids (FFAs) and, consequently, leads to activation of glucose oxidation, which is more economical in conditions of ischemia. An important feature of the action of Mildronate, distinguishing it from other drugs affecting myocardial metabolism, is the absence of accumulation of underoxidized fatty acids inside mitochondria, increasing NO production [19]. Our studies have confirmed the antihypoxic activity of Mildronate in PH [16]. The endothelioprotective effect of Mildronate has not been established by our studies and the present work. Course administration of Mildronate to rats after PH resulted in increased expression of various forms of glutathione peroxidase, which is consistent with other studies on its antioxidative activity [11,55].

However, this alone is insufficient to exert a protective effect on the cardiovascular system. In this study, we did not observe a significant positive effect of Mildronate on the NO system parameters in the myocardium of animals that underwent PH. L-arginine is a common substrate for NO and polyamines (putrescine, spermine, and spermidine). NO and polyamines play important roles in reproduction, embryogenesis, reducing neonatal mortality, and embryonic angiogenesis. NO regulates gene expression and protein synthesis, and facilitates the proliferation, growth, and differentiation of the fetus [56]. Currently, research is underway, and initial results have been obtained regarding the use of L-arginine in neonatology as a hypoxic and endothelial-targeting agent [13]. Our presented results, along with those from our previous studies, demonstrated the low effectiveness of L-arginine in experimental prenatal hypoxia [12,16]. Presumably, the NO formed from administered L-arginine becomes a target for ROS in the context of thiol compound deficiency, thereby failing to exert a protective effect [12,54]. A certain positive effect of L-arginine on molecular indices of endothelial dysfunction in the heart of rats after PH has been revealed. The weaker effect of L-arginine in comparison with Angiolin and Thiotriazoline can be explained from the point of view of NO life duration under ischemia and hypoxia accompanied by oxidative stress (Figure 4). “Newborn” NO immediately runs the risk of being ‘bitten’ by superoxydradicals [11,57] and converted into the sinister peroxynitrite [58,59,60].

Only combinations of L-arginine with SH-group donators or antioxidants can enhance its NO-modulating activity [54].

**Figure 4 pharmaceuticals-18-00106-f004:**
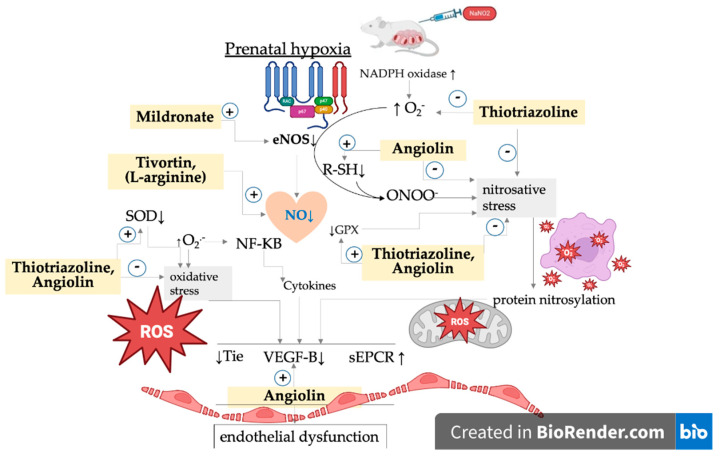
Hypothetical scheme of the mechanism of endothelioprotective action of NO modulators—arginine, Mildronate, Thiotriazoline, and Angiolin. The result of our research is the justification for the use of modulators of the nitric oxide (NO) system in the myocardium—Angiolin ((S)-2,6-diaminohexanoic acid 3-methyl-1,2,4-triazolyl-5-thioacetate), Thiotriazoline (morpholino-thiazotate), Mildronate, and L-arginine—with different mechanisms of action on NO concentration after prenatal hypoxia (PH). By regulating NO levels in the myocardium through various mechanisms, it is possible to interrupt NO-dependent mechanisms of endothelial dysfunction and cardiodestruction following PH. We have shown that the pharmacological level of NO and its bioavailability can be increased by (1) stimulating NO synthesis, for example, through therapy with the eNOS substrate L-arginine and the eNOS cofactor Mildronate, as well as directly protecting NO and even the eNOS protein from reactive oxygen species (ROS) using thiol-based antioxidants such as Thiotriazoline and Angiolin. We demonstrated the ability of all studied NO modulators to affect endothelial dysfunction markers in the myocardium after PH to varying degrees. Moderate effects of L-arginine and Mildronate were observed, which is related to the rapid loss of endogenous NO during oxidative stress and the low level of antioxidant protection in the myocardium after PH. Thiotriazoline and Angiolin were particularly promising in terms of effectiveness, as they protected NO from ROS, increased its bioavailability, prolonged its half-life, and optimized its use by various cellular systems aimed at overcoming endothelial dysfunction. They also protected eNOS from oxidative modification and loss of activity. Thiotriazoline and, especially, Angiolin exhibited both direct (formation of complexes) and indirect (increasing SOD expression) protective effects on NO from ROS. Through the NO-dependent mechanism, Angiolin and Thiotriazoline increased the nuclear density of endothelial cells in the myocardial vessels, normalized the expression of marker proteins such as sEPCR, Tie-2, and VEGF, and enhanced antioxidant defense tools.

## 4. Materials and Methods

### 4.1. Animal Characteristics

We employed fifty white female rats and ten males, each weighing between 220 and 240 g and about half a year old, sourced from the vivarium of the Institute of Pharmacology and Toxicology, National Medical Academy of Ukraine. The rats were kept in typical vivarium settings, which included a 20–25 °C temperature range, a 50–55% humidity level, a regular light cycle, and unlimited access to food and water suitable for their species. The “European Applicable Protection of Vertebrate Animals used for Experimental and Scientific Purposes” and the rules governing the collection of animals for biomedical research (Strasbourg, 1986, as revised in 1998) were followed in all manipulations. The Zaporizhzhia State Medical University Commission on Bioethics granted ethical permission for the study (protocol No. 33, dated 26 June 2021).

### 4.2. Experimental Model

In order to create a model based on nitrite, we caused chronic hypoxia, which significantly alters the histology, morphology, and metabolism of the progeny’s heart tissue [61,62]. Adult males were paired with females at a ratio of 2:4, and the first day of pregnancy was determined by the presence of spermatozoa in the vaginal smear. Daily intraperitoneal injections of a sodium nitrite solution at a concentration of 50 mg/kg were used to induce moderate hypoxia between days 16 and 21 of pregnancy [21]. An equivalent volume of physiological saline was given to control females. The offspring were separated into the subsequent groups: healthy rats from females undergoing normal pregnancies; a control group of pups that underwent PH and were administered physiological saline (days 1 to 30); 4 experimental groups of hypoxia-exposed pups that were treated daily with various drugs from postnatal days 1 to 30. Some of the pups were removed from the experiment on the 30th day, immediately after the completion of pharmacological agent administration, while others were removed 60 days after birth (30 days following treatment). The doses of L-arginine and Mildronate were sourced from the open literature. The doses of Thiotriazoline and Angiolin were determined experimentally, and these data are included in the DCT report.

### 4.3. Rationale for the Chosen Medications and Their Attributes

We chose treatments known to impact the NO system based on experimental evidence:The intact group consisted of rats born from females with basic pregnancies and received a physiological solution.The control group included rats born after experiencing intrauterine hypoxia and also received a physiological solution.Thiotriazoline, also known as morpholinium-3-methyl-1,2,4-triazolyl-5-thioacetic acid (2.5% injection solution, “Arterium”, Ukraine), is an antioxidant and metabolitotropic cardioprotector that is injected intraperitoneally at a dose of 50 mg/kg [63].Angiolin, additionally known as [S]-2,6-diaminohexane acid 3-methyl-1,2,4-triazolyl-5-thioacecate (substance, RPA “Farmatron”, Ukraine) is an endothelium-protective, anti-ischemic injection given intraperitoneally at a dose of 50 mg/kg [64].L-arginine (42% injection solution in vial, Tivortin, Yuria-pharm, Ukraine) is an NO precursor; to decrease ischemia-related nitroxidergic system disruptions, it is given intraperitoneally at a dose of 200 mg/kg [65].As a metabolitotropic drug, Mildronate (2-(2-carboxyethyl)-1,1,1-trimethylhydrazinium) (10% injectable solution in ampoules, Grindex (Latvia)) is injected intraperitoneally at a dose of 100 mg/kg [66].

### 4.4. Anesthesia

On days 30 and 60 of the trial, rats were put to sleep using thiopental anesthesia (40 mg/kg). For additional research, blood samples were taken from the celiac artery.

### 4.5. Biological Material Preparation

The heart was washed with a 1:10 dilution of cooled 0.15 M KCl solution and kept at 4 °C. After removing excess fat, connective tissue, blood vessels, and clots, the heart was rinsed with a 1:10 dilution of 0.15 M KCl solution at 4 °C. Utilizing a WT500 torsion balance (manufactured in Moscow, Russia), 100 milligrams of heart tissue was meticulously weighed after being previously ground into a fine powder using liquid nitrogen. Next, 10.0 mL of a medium kept at 2 °C was thoroughly mixed with the pulverized tissue. The concentration of the following ingredients in millimoles per liter (mmol/L) was adjusted to pH 7.4: 250 mmol/L of sucrose, 20 mmol/L of Tris-HCl buffer, and 1 mmol/L of EDTA. Large cell fragments were then extracted from the homogenate by pre-centrifuging it in a Sigma 3–30 k chilled centrifuge (Osterode am Harz, Germany) for 7 min at 1000× *g* at +4 °C. The resultant supernatant was carefully collected and put through a second centrifugation process using the same Sigma 3–30 k refrigerated centrifuge (Germany) for 20 min at 17,000× *g* at +4 °C. Following this procedure, the supernatant was collected and refrigerated at −80 °C. Subsequent to resuspension, the thick mitochondrial precipitate was employed for additional research. The apical part of the heart was placed in Bouin’s fixative for 24 h. After the standard tissue dehydration procedure and impregnation with chloroform and paraffin, the myocardium was embedded in Paraplast (MkCormick, Cockeysville, MD, USA). Serial histological sections, 5 μm thick, were prepared using a Microm-325 rotational microtome (Microm Corp., Munich, Germany). After treatment with xylene and ethanol, the sections were used for real-time PCR analysis and morphometric studies.

### 4.6. Immunoenzymatic Assay

The soluble endothelial protein C receptor (sEPCR) was measured in the cytosolic homogenate of the heart using a solid-phase enzyme-linked immunosorbent assay (ELISA) sandwich method. The assay was performed with the rat soluble endothelial protein C receptor (sEPCR) ELISA Kit, Catalog #MBS265381 from MyBioSource, Inc. (San Diego, CA, USA), following the provided instructions.

The Tie2 tyrosine kinase was also determined in the cytosolic homogenate of the heart using a solid-phase ELISA sandwich method. The assay was conducted with the Rat Tie2 (Rat Tek Tyrosine Kinase) Endothelial ELISA Kit, Catalog #MBS036226 from MyBioSource, Inc. (USA), in accordance with the instructions.

Vascular endothelial growth factor B (VEGF-B) was determined in the cytosol of heart homogenate using a solid-phase sandwich ELISA method, Rat Vascular Endothelial Growth Factor B (VEGF-B) ELISA Kit, Catalog # MBS269676 MyBioSource, Inc. (USA), according to the instructions.

SOD1/Cu-Zn SOD was determined in the cytosol of heart homogenate using a solid-phase sandwich ELISA method, Rat Superoxide dismutase [Cu-Zn] ELISA Kit, Catalog # MBS761294 MyBioSource, Inc. (USA), according to the instructions.

Glutathione peroxidase 4 (phospholipid hydroperoxidase) (GPX4) was determined in the cytosol of heart homogenate using a solid-phase sandwich ELISA method, Rat Phospholipid hydroperoxide glutathione peroxidase, mitochondrial, GPX4 ELISA Kit, Catalog # MBS934198 MyBioSource, Inc. (USA), according to the instructions.

Glutathione peroxidase 1 (GPX1) was determined in the cytosol of heart homogenate using a solid-phase sandwich ELISA method, Rat Glutathione Peroxidase 1 ELISA Kit, Catalog # MBS3809062 MyBioSource, Inc. (USA), according to the instructions. All studies were conducted using a plate enzyme immunoassay analyzer (SIRIO S, Ravenna, Italy).

### 4.7. Polymerase Chain Reaction in Real Time

The study of mRNA expression levels for VEGF and VEGF-B was conducted in the apical part of the myocardium. Sample preparation was performed as described in Section 4.5. The molecular–genetic study included several stages. Tissue samples were deparaffinized by incubation in two consecutive xylene baths for 5 min each, followed by two sequential baths of 100% ethanol for 5 min each. After deparaffinization and centrifugation, the precipitate was air-dried to remove ethanol residues.

Total RNA extraction from rat tissues was performed using the “Trizol RNA Prep 100” kit (“IZOGEN,” Moscow, Russia), which includes the following reagents: Trizol reagent and ExtraGene E. RNA was isolated according to the kit’s protocol. For reverse transcription (cDNA synthesis), the “Reagent Kit for Reverse Transcription (RT-1)” (“SINTOL”, Moscow) was used. Preparation and execution of the reaction were carried out according to the kit’s protocol.

#### Real-Time Polymerase Chain Reaction (RT-PCR)

To determine the expression levels of the studied genes, the CFX96™ Real-Time PCR Detection System (“Bio-Rad Laboratories, Inc.”, Hercules, CA, USA) and the SYBR Green R-402 PCR reagent kit (“Sintol”, Russia) were used. The final reaction mix for amplification included SYBR Green dye, SynTaq DNA polymerase with antibody-inhibited enzyme activity, 0.2 μL each of specific forward and reverse primer, dNTPs (deoxynucleoside triphosphates), and 1 μL of cDNA template. The reaction mix was brought to a total volume of 25 μL by adding deionized water.

Specific primer pairs (5′-3′) for analyzing the target and reference genes were designed using PrimerBlast software (July 2024, https://www.ncbi.nlm.nih.gov/tools/primer-blast/) and manufactured by ThermoScientific, USA. Amplification conditions were as follows: initial denaturation at 95 °C for 10 min, followed by 50 cycles of denaturation at 95 °C for 15 s, primer annealing at 58–63 °C for 30 s, and elongation at 72 °C for 30 s. Fluorescence intensity was automatically recorded at the end of the elongation phase of each cycle using the SybrGreen channel.

The beta-actin (Actb) gene was used as a reference gene to determine the relative changes in the expression levels of the studied genes [67,68].

### 4.8. Morphometric Studies

Histological sections were stained with hematoxylin and eosin and embedded in the polymer medium EUKITT (O. Kindler GmbH, Freiburg, Germany) for microscopy. The cross-sectional area of endothelial cell nuclei in arterioles with diameters of 30–50 µm was measured in histological samples at ×400 magnification. The study was conducted using a Carl Zeiss Axio Scope.A1 microscope (Germany) paired with a Jenoptik Progres Gryphax^®^ Subra digital camera (Jena, Germany) and GRYPHAX software (version 2.2.0.1234). Measurements were taken in 10 fields of view using the VideoTest-Morphology software version 5.2.0.158.

The cross-sectional area of endothelial cell nuclei in arterioles with a diameter of 30–50 μm and the area of cardiomyocyte nuclei were determined on histological slides stained with hematoxylin and eosin at ×400 magnification. The studies were performed using an Axio Scope A1 microscope (Carl Zeiss, Oberkochen, Germany) equipped with a Jenoptik Progress Gryphax^®^ SUBRA series camera (Germany).

In each case, a morphometric analysis of the structural components of the myocardium was conducted in 10 fields of view using the VideoTest Morphology software, version 5.2.0.158. Calibration was performed prior to the study, corresponding to the working magnification of the microscope used to obtain the histological images. The next step involved isolating object masks, distinguished by optical density parameters (Figure 5). Subsequently, the areas of the objects were measured automatically, excluding internal cavities.

The analyzed parameters were presented in a table format (Figure 6) and exported to Excel for further graphical processing. Statistical analysis of the obtained data was also performed using the Statistica^®^ for Windows 13.0 software package (StatSoft Inc., Tulsa, OK, USA, license no. JPZ804I382130ARCN10-J).

**Figure 5 pharmaceuticals-18-00106-f005:**
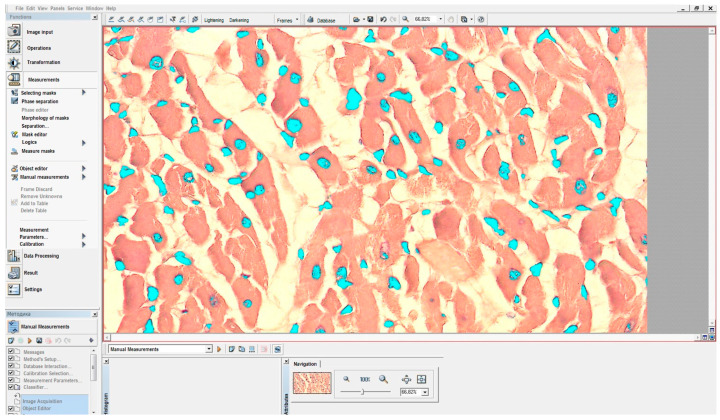
Highlighted cardiomyocyte nuclei (object masks) based on the gradient of optical density.

### 4.9. Statistical Analysis

Experimental data were statistically analyzed using “Statistica^®^ for Windows 6.0” (StatSoft Inc., Tulsa, OK, USA, AXXR712D833214FAN5), “SPSS16.0”, and “Microsoft Office Excel 2010” software. Prior to statistical tests, we checked the results for normality (Shapiro–Wilk and Kolmogorov–Smirnov tests). In the normal distribution, intergroup differences were considered statistically significant based on the parametric Student’s *t*-test. If the distribution was not normal, the comparative analysis was conducted using the non-parametric Mann–Whitney U-test. To compare independent variables in more than two selections, we applied ANOVA dispersion analysis for the normal distribution and the Kruskal–Wallis test for the non-normal distribution. To analyze correlations between parameters, we used correlation analysis based on the Pearson or Spearman correlation coefficient. For all types of analysis, the differences were considered statistically significant at *p* < 0.05 (95%).

## 5. Conclusions

We have obtained convincing results indicating that the modeled PH leads to significant disorders in the cardiovascular system of offspring (1- and 2-month-old rats). In the myocardium of rats that underwent PH, an increase in the marker of endothelial dysfunction—sEPCR—was detected against a background of decreased Tie-2 and VEGF-B, which perform protective functions, alongside antioxidant deficiency reduction as well as Cu/ZnSOD and GPX. Our results experimentally substantiate the necessity for early postnatal cardio- and endothelioprotection using NO modulators, considering the role of NO-dependent mechanisms in the pathogenesis of cardiovascular system disorders in newborns after PH. We have shown that only two representatives of this group, Angiolin and Tiotriazoline, are capable of exerting a complete effect on the indicators of endothelial dysfunction after PH (with a decrease in sEPCR against an increase in Tie-2, VEGF-B, and Cu/ZnSOD, GPX), which perform protective functions and antioxidative functions. Based on the conducted research, the feasibility of further preclinical studies of Angiolin as a promising means of cardioprotection after PH has been experimentally justified. Additionally, the results obtained support the potential for conducting further preclinical and clinical studies of Tiotriazoline (as an approved medication) as a treatment for cardiovascular system pathologies following intrauterine hypoxia.

Prospects for Future Research. In the future, we plan to conduct a more detailed study of the endothelial-protective mechanism of Thiotriazoline and Angiolin in animals following intrauterine hypoxia. This will involve examining their effects on the density of proliferating endothelial cells, RNA content in endothelial cell nuclei, and the density of VEGF- and eNOS-positive cells in various types of vessels. These studies will employ methods such as electron microscopy, light microscopy, and immunohistochemistry.

## Figures and Tables

**Figure 1 pharmaceuticals-18-00106-f001:**
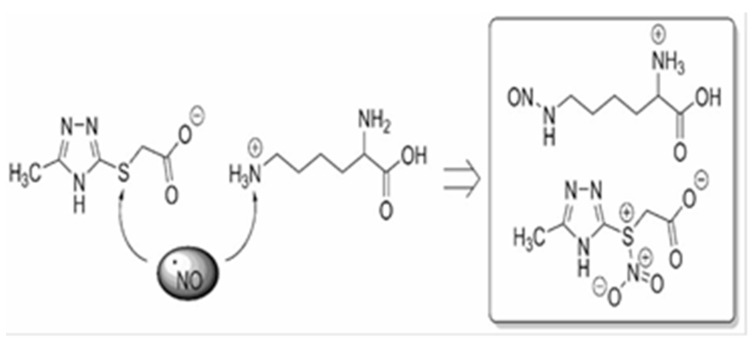
Hypothetical mechanism of (S)-2,6-diaminohexanoic acid 3-methyl-1,2,4-triazolyl-5-thioacetate (Angiolin) interaction with NO. The NO scavenger properties of Angiolin are realized through the reactivity of both the cationic and anionic parts of the molecule (S)-2,6-diaminohexanoic acid 3-methyl-1,2,4-triazolyl-5-thioacetate. Specifically, L-lysine interacts with NO via the ε-amino group, resulting in the corresponding N-nitrosated derivative. Simultaneously, the anionic part of the molecule (S)-2,6-diaminohexanoic acid 3-methyl-1,2,4-triazolyl-5-thioacetate (Angiolin) likely forms S-nitro derivatives, as described elsewhere. The NO scavenger properties of both the anionic and cationic parts of (S)-2,6-diaminohexanoic acid 3-methyl-1,2,4-triazolyl-5-thioacetate appear to be synergistic, which accounts for the outstanding effect observed with the investigated Angiolin. The mechanism of interaction between the Angiolin molecule and NO may involve the transfer of an electron from the highest occupied molecular orbital of the “spin trap” to the lowest unoccupied molecular orbital of the radical, forming a more stable radical complex. We performed calculations of quantum-mechanical energy descriptors for the frontier molecular orbitals, the energy of the highest occupied molecular orbital (EHOMO) and the energy of the lowest unoccupied molecular orbital (ELUMO), using the WinMopac software package (version 7.2, descriptors HOMOEnergy, LUMOEnergy, AM1 method, with settings: Calculation = SinglePoint, Wavefunction = ClosedShell (RHF)). Additionally, the following characteristics were calculated: the energy gap (the difference between the HOMO and LUMO energies); absolute hardness using the formula η = − (EHOMO − ELUMO)/2; and absolute electronegativity using the formula χ0 = − (EHOMO + ELUMO)/2. The quantum chemical parameters of the Angiolin molecule correlate with our earlier studies and show that the EHOMO parameter (HOMOEnergy descriptor) has the greatest influence on the oxidative stress marker nitrotyrosine, which is directly proportional to the concentration of the NO-peroxynitrite degradation product. The interaction mechanism of the Angiolin molecule with NO can be realized through the transfer of an electron from the highest occupied molecular orbital of the “spin trap” to the lowest unoccupied molecular orbital of the radical, forming a more stable radical complex. Thus, Angiolin may act as an NO transporter molecule, potentially playing a crucial role in the endothelial defense mechanism. The obtained results are consistent with computational findings [51].

**Figure 6 pharmaceuticals-18-00106-f006:**
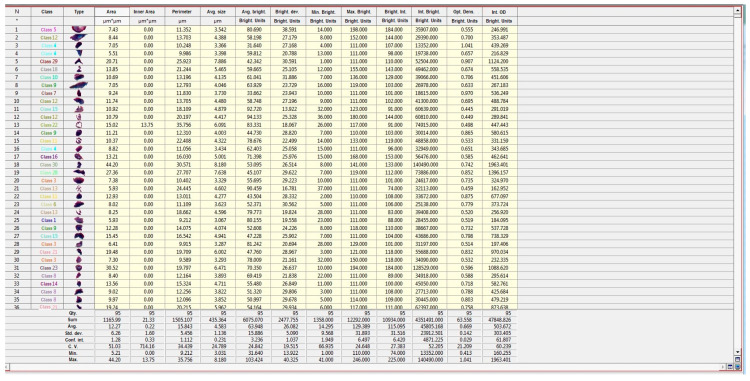
Results of quantitative measurements obtained using the VideoTest Morphology 5.2.0.158 software. Statistical analysis of the data was performed using Statistica^®^ for Windows 13.0 (StatSoft Inc., USA, license no. JPZ804I382130ARCN10-J), and graphical data processing was completed using Microsoft Excel.

**Table 1 pharmaceuticals-18-00106-t001:** System parameters of cytosolic fraction in 1-month-old rats after prenatal hypoxia and treatment.

Experimental Groups	sEPCR, pg/mL	Tie-2, pg/mL	VEGF-B, pg/mL	Cu/ZnSOD, pg/mL	GPX1, pg/mL	GPX4, pg/mL
Intact (rats born from rats with normal pregnancies) (*n* = 10)	22.5 ± 0.411	17.7 ± 0.348	44.7 ± 1.012	87.7 ± 1.802	43.3 ± 1.044	67.8 ± 1.676
PH (rats with prenatal hypoxia) (control) (*n* = 10)	43.2 ± 1.360 ^1^	10.2 ± 0.275 ^1^	32.1 ± 1.012 ^1^	63.5 ± 1.360 ^1^	21.1 ± 0.538 ^1^	34.2 ± 0.537 ^1^
PH + L-arginine (*n* = 10)	38.0 ± 0.854 ^1,^*	14.2 ± 0.348 ^1,^*	34.7 ± 1.486 ^1^	62.7 ± 1.739 ^1^	22.8 ± 0.696 ^1^	38.3 ± 1.328 ^1,^*
PH + Thiotriazoline (*n* = 10)	33.5 ± 1.012 ^1,^*	12.7 ± 0.316 ^1,^*	36.8 ± 1.170 ^1,^*	77.8 ± 1.961 ^1,^*	38.8 ± 0.696 ^1,^*	57.7 ± 0.949 ^1,^*
PH + Angiolin (*n* = 10)	28.2 ± 0.538 ^1,^*	16.4 ± 0.380 ^1,^*	47.8 ± 0.885 ^1,^*	79.7 ± 1.676 ^1,^*	40.7 ± 1.012 *	62.8 ± 1.803 *
PH + Meldonium (*n* = 10)	40.5 ± 2.119 ^1^	11.0 ± 0.231 ^1,^*	31.1 ± 1.170 ^1^	65.2 ± 1.961 ^1^	22.7 ± 0.348 ^1^	37.3 ± 0.601 ^1^*

Notes: ^1^—*p* ≤ 0.05 in relation to the intact group of animals; *—*p* ≤ 0.05 in relation to the control group of animals.

**Table 2 pharmaceuticals-18-00106-t002:** Cytosolic fraction parameters in 2-month-old rats after prenatal hypoxia and treatment.

Experimental Groups	sEPCR, pg/mL	Tie-2, pg/mL	VEGF-B, pg/mL	Cu/ZnSOD, pg/mL	GPX1, pg/mL	GPX4, pg/mL
Intact (rats born from rats with normal pregnancies) (*n* = 10)	21.2 ± 0.348	18.2 ± 0.253	48.8 ± 1.012	91.9 ± 2.308	46.4 ± 0.664	72.4 ± 1.676
PH (rats with prenatal hypoxia) (control) (*n* = 10)	45.4 ± 0.727 ^1^	11.3 ± 0.221 ^1^	31.6 ± 0.696 ^1^	62.8 ± 1.581 ^1^	21.2 ± 0.949 ^1^	37.8 ± 0.569 ^1^
PH + L-arginine (*n* = 10)	35.2 ± 0.537 ^1,^*	15.2 ± 0.348 ^1,^*	32.7 ± 0.854 ^1^	66.7 ± 1.328 ^1^	24.3 ± 0.569 ^1,^*	39.4 ± 0.443 ^1^
PH + Thiotriazoline (*n* = 10)	32.2 ± 0.569 ^1,^*	15.7 ± 0.243 ^1,^*	37.8 ± 1.075 ^1,^*	78.7 ± 1.992 ^1,^*	42.6 ± 0.791 ^1,^*	68.7 ± 1.360 ^1,^*
PH + Angiolin (*n* = 10)	21.2 ± 0.632 *	18.4 ± 0.379 *	52.8 ± 1.202 ^1,^*	88.7 ± 2.625 *	48.8 ± 1.075 *	77.8 ± 1.834 ^1,^*
PH + Meldonium (*n* = 10)	44.9 ± 1.676 ^1^	10.4 ± 0.127 ^1,^*	34.7 ± 0.601 ^1,^*	64.4 ± 1.391 ^1^	27.4 ± 0.601 ^1,^*	42.5 ± 1.518 ^1,^*

Notes: ^1^—*p* ≤ 0.05 in relation to the intact group of animals; *—*p* ≤ 0.05 in relation to the control group of animals.

**Table 3 pharmaceuticals-18-00106-t003:** VEGF mRNA and VEGF-B mRNA expression in myocardial tissues of 1-month-old rats after prenatal hypoxia and treatment.

Experimental Groups	VEGF mRNA, a.u.	VEGF-B mRNA, a.u.
Intact (rats born from rats with normal pregnancies) (*n* = 10)	1.000 ± 0.0017	1.000 ± 0.009
PH (rats with prenatal hypoxia) (control) (*n* = 10)	0.34 ± 0.00012 ^1^	0.170 ± 0.00011 ^1^
PH + L-arginine (*n* = 10)	0.31 ± 0.00012 ^1^	0.18 ± 0.0022 ^1^
PH + Thiotriazoline (*n* = 10)	0.812 ± 0.0055 ^1,^*	1.17 ± 0.0011 ^1,^*
PH + Angiolin (*n* = 10)	1.731 ± 0.0121 ^1,^*	2.13 ± 0.0021 ^1,^*
PH + Meldonium (*n* = 10)	0.331 ± 0.0021 ^1^	0.18 ± 0.0022 ^1^

Notes: ^1^—*p* ≤ 0.05 in relation to the intact group of animals; *—*p* ≤ 0.05 in relation to the control group of animals.

**Table 4 pharmaceuticals-18-00106-t004:** Expression of VEGF mRNA and VEGF-B mRNA in myocardial tissues of 2-month-old rats after prenatal hypoxia and treatment.

Experimental Groups	VEGF mRNA, a.u.	VEGF-B mRNA, a.u.
Intact (rats born from rats with normal pregnancies) (*n* = 10)	1.000 ± 0.0012	1.000 ± 0.0015
PH (rats with prenatal hypoxia) (control) (*n* = 10)	0.30 ± 0.00014 ^1^	0.156 ± 0.0003 ^1^
PH + L-arginine (*n* = 10)	0.31 ± 0.00021 ^1^	0.161 ± 0.0014 *
PH + Thiotriazoline (*n* = 10)	1.17 ± 0.0011 ^1,^*	2.00 ± 0.0021 ^1,^*
PH + Angiolin (*n* = 10)	1.87 ± 0.0023 ^1,^*	3.05 ± 0.0023 ^1,^*
PH + Meldonium (*n* = 10)	0.31 ± 0.0011 ^1^	0.143 ± 0.0001 ^1^

Notes: ^1^—*p* ≤ 0.05 in relation to the intact group of animals; *—*p* ≤ 0.05 in relation to the control group of animals.

**Table 5 pharmaceuticals-18-00106-t005:** Cross-sectional area of endothelial cell nuclei in the myocardial apex of 1- and 2-month-old rats after prenatal hypoxia and treatment.

Experimental Groups	Cross-Sectional Area of Endothelial Cell Nuclei, μm^2^	
	1-Month-Old Rats	2-Month-Old Rats
Intact (rats born from rats with normal pregnancies) (*n* = 10)	15.23 ± 2.34	16.17 ± 2.65
PH (rats with prenatal hypoxia) (control) (*n* = 10)	8.77 ± 0.76 ^1^	9.15 ± 0.67 ^1^
PH + L-arginine (*n* = 10)	8.98 ± 1.23 ^1^	11.71 ± 1.10 ^1,^*
PH + Thiotriazoline (*n* = 10)	12.34 ± 1.24 ^1,^*	14.82 ± 1.58 ^1,^*
PH + Angiolin (*n* = 10)	14.11 ± 2.35 *	15.61 ± 1.76 *
PH + Meldonium (*n* = 10)	8.34 ± 0.714 ^1^	8.77 ± 1.11 ^1^

Notes: ^1^—*p* ≤ 0.05 in relation to the intact group of animals; *—*p* ≤ 0.05 in relation to the control group of animals.

## Data Availability

The manuscript contains the original contributions made during the study; further inquiries can be directed to the corresponding authors.

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
