# Peer review of "Possibility of Using NO Modulators for Pharmacocorrection of Endothelial Dysfunction After Prenatal Hypoxia"

_pharmaceuticals, 2025, doi:10.3390/ph18010106_

Round 1
Reviewer 1 Report
Comments and Suggestions for Authors
In this manuscript, the authors aim to evaluate the effect of different pharmacological modulators of the NO system with different mechanisms of action on some molecular markers of endothelial dysfunction in the early postnatal period after intrauterine hypoxia.
Although the topic is very interesting from a clinical point of view, the work has some weaknesses, which are described below.
1. The use of nitrite as a method of inducing prenatal hypoxia is the major weakness of the model used in this study. It has been reported that nitrite can lead to the development of hypoxia due to decreased oxygen delivery to tissues and impaired oxygen utilization due to inhibition of mitochondrial respiration. However, this chemical hypoxia may have many undesirable effects due to the toxicity of nitrite as described [Evidence on Developmental and Reproductive Toxicity of Sodium Nitrite. Reproductive and Cancer Hazard Assessment Sec-677 tion (RCHAS) Office of Environmental Health Hazard Assessment (OEHHA) California Environmental Protection Agency 678 (CAL/EPA) DRAFT. USA. 3 March 2000].
On the other hand, nitrite anion is involved in many physiological and pathological processes in the body. Its special function is that it can be easily reduced to NO by the action of nitrite reductase systems. In this respect, nitrite is considered an alternative source of NO in normal physiological conditions and in pathological endothelial dysfunction.
These two functions of nitrite are in conflict: on the one hand, nitrite causes hypoxia in the pregnant mother and, at the same time, nitrite interferes with endothelial function, whose changes in the offspring are studied here and whose postnatal epigenetic effects are unknown.
In summary, it would be necessary to validate the results of this model with a hypoxic treatment without chemical agents, by exposing the mothers to a hypoxic environment during at least part of the pregnancy.
2. The methodology described is confusing and needs further explanation. While ELISAs are described in detail, relevant information on PCR is omitted. It is not explained which part of the tissue is used for RNA extraction, how the mRNA is extracted, which primers are used, and the method for calculating the RNA expression is omitted.
The same applies to the morphometric measurements, which are only vaguely explained and no image is shown.
Overall, this study would be difficult to replicate due to the lack of methodological explanations.
Finally, it is striking that no vessel was used in a study of endothelial dysfunction.
3. The results, consisting only of tables, are very descriptive and the discussion is over-interpreted, although Figure 2 is highly appreciated for clarification.
4. There is an excessive number of Russian language references. For an English-language journal such as Pharmaceuticals, this is a handicap as readers cannot access the content of these references
Author Response
We thank the reviewer for the constructive feedback and comments:
Answer1. A significant challenge in all experimental medicine is the development or selection of an appropriate experimental pathology model (if such models already exist). This is a major concern for many researchers, particularly pharmacologists, because the choice of model determines the accuracy of preclinical assessments of new drugs, dose calculations, and the analysis and determination of the primary mechanism of action. In global preclinical research practices, as well as in pathophysiology and pathochemistry, there are several models of prenatal hypoxia (PH). There are many options to list them all. However, we note that we have chosen the least traumatic and most clinically relevant model for offspring. This model replicates almost all the electrophysiological, morphological, and molecular-biochemical changes in the myocardium observed in newborns and older children who have experienced PH of varying severity. In the "Discussion" section, we have included references to sources supporting this.
Regarding the comments about the nitrite anion as an agent inducing PH, there may have been some misunderstanding. In the "Discussion" section, we provided clarifications, supported by references, explaining that this model specifically involves hypoxia induced by the nitrite anion, not its direct cytotoxic effect. As for the duality of nitric oxide (NO)—its cytoprotective and cytotoxic effects—and the agents influencing the nitric oxide system, there are numerous reviews and original articles, including some of our own. These sources are also cited in the reference list.
Answer2. We now provided a more detailed description of the real-time PCR method and morphometry method
Answer3. In our study, to confirm the endothelioprotective effect of pharmacological agents, we used such an indicator as the density of endotheliocyte nuclei of small and medium-sized myocardial vessels, which makes it possible (together with molecular genetic markers) to assess the possible effect on the endothelium. We agree that there are more direct methods - e.g. to test with acetylcholine and others. However, we have included in this manuscript the study that makes it possible to evaluate the effect of NO modulators on specific markers of endothelial dysfunction.
Answer4. We also had vague doubts about references to Russian-language sources, but there were no similar studies available in English to refer to, and thus we had no choice since the primary experimental material on these pharmacological agents was published in Russian. We believe that this will not be an issue for the readers, as these sources are freely available online and there are available technologies today that make it easy and possible to translate the text of the original sources, when needed.
Reviewer 2 Report
Comments and Suggestions for Authors
The present manuscript describes the in vivo evaluation of some compounds that could affect/restore nitric oxide bioavailability in a PH model as potential therapeutic agents for this condition. I have some serious concerns about this work.
· The purpose of the study is not clear. It seems that from one hand Authors aim to restore eNOS activity by scavenging the nitro-oxidative stress by thiatriazoline and angiolin, and from the other they want to stimulate the NO production by its precursor L-arginine or a stimulator like mildronate. It should be better explained the rationale for the selection of the studied compounds
· Results. This section is not clearly presented. Tables are not explicative; graphs and both PCR and western blot analysis should be reported and properly discussed.
· The discussion section is too long, especially from line 377 to 500, while the part regarding the obtained results leaves room for different doubts. Angiolin is supposed to be a scavenger and a NO donor at the same time…why should it be beneficial? Authors say that “Angiolin normalizes eNOS/iNOS expression “ but they don’t prove this. Also the discussion of thiatriazoline biological profile is contradictory, being this molecule considered as a NO scavenger and a NOS stimulator at the same time (“Thiotriazoline, a drug registered in many countries as a methiabolitotropic cardio-protective agent, also exhibits NO scavenger properties…..Thiotriazoline can increase endothelioprotective properties of L-arginine by increasing NO bioavailabiliity [56,57,58]” ) Also the discussion of the L-arginine expected vs obtained results is confusing.
Minor points:
The Introduction section should be improved. Pag. 2 line 66: explain NO abbreviation. In general, explain how the cited factors are changed. Pag 2 line 72:” superoxydradicals”. What is intended? Please introduce molecular formula. The same for “peroxynitrite”. Pag. 2 lines 91-93. Add a figure with the molecular structure of Thiatriazolin. Add references about its capability to scavenge NO and ROS. The same comments are fot Angiolin (pag.2 and 3 lines 98-103)
Conclusions:
In my opinion the present manuscript has serious weaknesses in the methodological part and conclusions are not supported by the obtained data. For these reasons I do not recommend its publication in Pharmaceuticals.
Comments on the Quality of English LanguageEnglish grammar and style should be revised.
Author Response
We thank the reviewer for the constructive feedback and comments. We revised the manuscript accordingly, however, we respectfully disagree with some of the suggestions and would like to provide our clarifications below.
Answer1. The authors do not claim to make groundbreaking scientific discoveries, they rather aim to optimize the management of cardiovascular complications following prenatal hypoxia (PH). The authors chose nitric oxide (NO) as the pharmacological target because its deficiency associated with reduced eNOS expression and activity, and its “destruction” by reactive oxygen species (ROS) leads to severe cardiac consequences post-PH. Through the text, including in the Discussion section, authors clarified the rationale and actions aimed at increasing NO levels and its bioavailability using pharmacological agents with different mechanisms of action. The authors have now expanded the explanation regarding the choice of agents targeting the NO system.
Answer2. The authors believe that the stated aim is sufficiently concise in the revised manuscript. However, the authors are open to further rephrasing or revising it, if needed.
Answer3. The Reviewer's comments on the description of the results are somewhat unclear for the authors. The authors did not use western blot in the present study. The "Results" section is presented in a classical format with data provided in tables.
Answer4. In the "Discussion" section, the authors aimed to review the continuity of their findings taking into consideration previous research and data from other scientists in this field as well. While it may seem somewhat lengthy, the authors believe that it holds explanatory value. Regarding the discussion of the effects of Angiolin and Thiotriazoline, the authors have now clarified and explained their action through the prism of their impact on the NO system, citing previously obtained experimental data—normalization of eNOS/iNOS expression (with references to primary sources), increased NO bioavailability (supported by references and an explanatory diagram), and protection of NO from ROS. The authors did not describe Angiolin as an NO donor.
Answer5. Regarding Thiotriazoline, there authors find no contradictions. The manuscript contains references to primary sources. There is also newly added information explaining Thiotriazoline's ability to increase NO bioavailability and influence eNOS activity. Two figures were now included to support this explanation. All of this is understandable through the antioxidant mechanism of Thiotriazoline's action.
Answer6. The authors find no particular ambiguity in explaining the action of L-arginine. The authors noted that it has attracted the attention of clinicians, with initial results of its use post-PH offering promise. However, the authors could not confirm overly optimistic claims about its effectiveness. There is now a statement included that clarifies this point.
Answer7. The article's style and grammar now have been revised. The structural formula of Angiolin is now presented in the diagram illustrating its interaction with NO, and a similar depiction was added for Thiotriazoline. The authors do not see the necessity of including structural formulas for all the molecules when the chemical names are already provided.